# A Study on the Stability and Antimicrobial Efficacy of a Newly Modeled Teat Dip Solution Containing Chlorhexidine

**DOI:** 10.3390/vetsci10080510

**Published:** 2023-08-08

**Authors:** Modestas Kybartas, Marius Virgailis, Modestas Ruzauskas, Irena Klimienė, Rita Šiugždinienė, Lina Merkevičienė, Žaneta Štreimikytė-Mockeliūnė, Raimundas Mockeliūnas

**Affiliations:** Microbiology and Virology Institute, Veterinary Academy, Lithuanian University of Health Sciences, 47181 Kaunas, Lithuania; marius.virgailis@lsmuni.lt (M.V.); modestas.ruzauskas@lsmuni.lt (M.R.); irena.klimiene@lsmuni.lt (I.K.); rita.siugzdiniene@lsmuni.lt (R.Š.); lina.merkeviciene@lsmuni.lt (L.M.); zaneta.streimikyte@lsmu.lt (Ž.Š.-M.); raimundas.mockeliunas@lsmuni.lt (R.M.)

**Keywords:** teat dip solution, chlorhexidine, antimicrobial efficacy, stability, viscosity, hydroxypropyl guar gum

## Abstract

**Simple Summary:**

Bovine mastitis is one of the most widespread cow diseases, which causes high economic losses. Local treatment using natural ingredients instead of the systematic use of antibiotics can decrease the development of antimicrobial resistance. The aim of the study was to determine the physico-chemical properties, stability and antimicrobial effect of a newly formulated biocide for post-milking udder hygiene containing a thickener made from hydroxypropyl guar gum, an antiseptic chlorhexidine digluconate and teat skin-friendly components, including glycerol, *Mentha Arvensis* herbal oil and *Aesculus hippocastanum* extract. The product remains stable and homogenous for at least 12 months. The product also has good antimicrobial properties against the main mastitis pathogens, including *Staphylococcus aureus*, *Streptococcus uberis*, *Escherichia coli*, *Candida albicans* and *Aspergillus niger*.

**Abstract:**

Despite much focus on mastitis as an endemic disease, clinical and subclinical mastitis remains an important problem for many herds. Reducing the usage of antibiotics for mastitis treatment allows the risks to be minimized related to the development of antimicrobial resistance and the excretion of antibiotics into the environment. The aim of the study was to determine the physico-chemical properties, stability and antimicrobial effect of a newly formulated biocide for post-milking udder hygiene containing a thickener made from hydroxypropyl guar gum, an antiseptic chlorhexidine digluconate and teat skin-friendly components including glycerol, *Mentha Arvensis* herbal oil and *Aesculus hippocastanum* extract. Hydroxypropyl guar gum was used as a thickener to provide the physical parameters and to retain the viscosity at 1438 mPa.s. The physical and chemical properties of the product, including the 12-month stability, were tested in long-term and accelerated stability studies. The product was effective against the primary mastitis pathogens, including *Staphylococcus aureus*, *Streptococcus uberis*, *Escherichia coli*, *Candida albicans* and *Aspergillus niger*.

## 1. Introduction

Despite the extensive focus on mastitis as an endemic disease, clinical and subclinical mastitis remains an important problem for many herds. The disinfection of teats after milking reduces the number of new mastitis infections (IMIs) by 50–70%, particularly those caused by contagious mastitis pathogens, and it achieves more variable effects for environmental pathogens [1]. Teat disinfection is a very integral part of the prevention of mastitis, which helps to reduce the risk of pathogens transmission between animals [2]. There is clear evidence that the incidence of new IMIs caused by a wide variety of pathogens was much lower after post-milking teat disinfection. Moreover, this inexpensive and simple technique ensured reduced bulk milk SCC during lactation and had fewer teat skin abnormalities compared with cows without disinfection [1,3]. For this reason, post-milking teat disinfection is included into many mastitis control programs.

Post-milking dip solutions should include certain features suitable for teat hygiene: they must have antimicrobial activity, quickly cover the teat and properly remain on the surface for a certain period; they must be the right pH in order not to irritate the skin; and they must perform a moisturizing effect to keep the skin soft, rather than cracked, and help any minor wounds to heal more quickly [4,5]. Therefore, antiseptics which are used for cow udders after milking may contain bactericidal and bacteriostatic substances. The use of different active substances in biocides provides a broad spectrum of disinfection. Polymers of guar or xanthan or guar gum provide thickening viscosity to biocides. As a result, the teat dip solution stays on the teat for the right amount of time [6,7,8,9].

The teat dip products should contain a disinfecting agent which is friendly to the skin of the teats and is not harmful to people working with animals. That agent is chlorhexidine, which is extensively utilized in veterinary medicine. Chlorhexidine can be used to disinfect the skin of animals; antiseptics of this type are widespread for the udder hygiene of cows before and after milking [6]. Thanks to using post-milking hygiene products containing chlorhexidine, according to the authors, the udder is protected from various inflammations and microorganisms that can enter into the teat canal. At high concentrations (>0.1%), CHX caused a leakage of all the main intracellular components out of the cell, resulting in a bactericidal (cell lysis and death) effect. At low concentrations (0.02–0.06%), CHX caused a displacement of Ca^2+^ and Mg^2+^ and losses of K^+^ from the cell wall, resulting in a bacteriostatic effect. Therefore, teat disinfection with chlorhexidine is an important tool in reducing the incidence of bovine mastitis [10].

Despite the biocidal properties, an important physical aspect is viscosity and thickness. After the dipping on the teat, skin creates a protective film that protects the teat canal from pathogens [11]. Viscosity can be adjusted by choosing different types of thickeners and their different concentrations. They also help maintain product stability and improve rheological properties [12]. In our opinion, the color of the teat dip is not of less importance, as the teat dip should be seen on the skin; i.e., the substance should contain a dye that gives a color to the product that acts as a control marker for the personnel and equipment. The efficient teat dip should be chosen taking into account its composition, the proportion of the materials and the combined stability and effectiveness. Biocidal products must maintain sufficient stability throughout their shelf life. The viscosity, pH, color, homogeneity stability, and odor intensity must remain.

The aim of this study was to determine the physico-chemical properties, stability and antimicrobial effect of a newly formulated biocide for post-milking udder hygiene containing a thickener made from hydroxypropyl guar gum, an antiseptic chlorhexidine digluconate and teat skin-friendly components, including glycerol, *Mentha Arvensis* herbal oil and *Aesculus hippocastanum* extract.

## 2. Materials and Methods

### 2.1. Place and Study Design

Investigations were carried out at the Institute of Microbiology and Virology, Lithuanian University of Health Sciences, as well as at the laboratory of Chelab S.R.L. in compliance with Good Laboratory Practices, Resana, Italy. The research was carried out in compliance with the Republic of Lithuania’s Animal Welfare and Protection Act (no. 108-2728; 2012, no. 122-6126). The study approval number was G2-227, which was issued by the Lithuanian State Food and Veterinary Service. The study design is presented in Table 1.

### 2.2. Development and Preparation of Teat Dip Solution

The solution formula was formulated by carefully selecting and adjusting the concentrations of the hydroxypropyl guar gum in order to achieve an appropriate viscosity and teat skin color after dipping. Chlorhexidine was used as the antimicrobial substance whereas plant extracts were included as antiseptic and skin-conditioning ingredients. All ingredients used for the development of the new model of teat dip solution are presented in Table 2.

The modeling of the initial experimental compositions used for the teat dip solution was carried out through an analogous method and by varying the amounts of the viscosity-regulating material and dye, with three experimental compositions being prepared. See Table 2 for the compositions. Hydroxypropyl guar gum was dispersed in purified water using an IKA ULTRA-TURRAX^®^ T 25 digital laboratory homogenizer (IKA-Werke GmbH & Co. KG, Staufen, Germany), and, if necessary, the pH of this solution was adjusted to 6.5–6.8 with citric acid. *Mentha arvensis* herb oil and glycerol were dissolved in isopropyl alcohol. *Aesculus hippocastanum* dry extract and dye were dissolved in purified water and filtered. Polysorbate 80 was dissolved in the remaining purified water, and the pH of this solution was adjusted with sodium hydroxide when needed to achieve an appropriate pH range of 5.8–6.1. A mixture of isopropyl alcohol, glycerol and *Mentha Arvensis* herb oil was gradually added to the hydroxypropyl guar gum solution. Thereafter, a mixture of *Aesculus hippocastanum* dry extract and dye was added, mixed and then polysorbate 80 was added. During the last stage of the mixing process, a 20% solution of chlorhexidine digluconate was added to the mixture, and the solution was mixed again using a EUROSTAR 20 digital laboratory stirrer (IKA-Werke GmbH & Co. KG, Germany) at a speed of 1250 rpm for 10 min to obtain a homogeneous mass. The equilibration of the modeled experimental compositions (EC) was carried out for 24 h at room temperature (15–25 °C).

### 2.3. In Vivo Testing

The in vivo study was carried out in order to evaluate the compliance of the simulated three experimental compositions with the provided requirements. The in vivo study was carried out in 2019–2021 at a dairy cow farm located in Siauliai district, Lithuania. The dairy farm houses 450 cows of the Lithuanian Red breed, with an average productivity of 10,500 kg per lactation. Cows are milked in the milking parlor twice a day. The in vivo study was carried out with 75 cows; i.e., three test groups of 25 cows each were formed to evaluate each experimental composition. After milking, cow teats were dipped immediately with EC solution up to 3/4 of the teat length. During the in vivo study, physical tests of the experimental compositions were performed: color, dripping immediately after dipping, formation of a drop on the teat end, teat covered with a film, evenness of teat coverage. The protocols record the values of the research indicators by visually applying a 5-point system. The evaluation criteria are presented in Table 3. The composition with the highest score was selected for further study.

### 2.4. Stability Testing

The stability evaluation was conducted in accordance with the guidelines for the stability testing of cosmetic products [13,14]. In the study conducted in an accelerated way, a regime of increased temperature was applied (2 months at +45 °C and 75 ± 5% relative humidity), more intense heating/cooling cycles (10 days at +4 °C; 10 days at +45 °C; 10 days at +4 °C; 10 days at +45 °C; 10 days at 15 °C, and 75 ± 5% relative humidity). The accelerated stability tests were performed twice: one day after manufacturing and after storage at the specified conditions. When testing a teat dip solution in the long-term study, the product samples were stored, and changes of the product characteristics were assessed after a day, after half a year, after 1 year, and after 2 years, under the recommended study conditions (25 ± 2 °C temperature and 60 ± 5% relative humidity). A “Binder KBF” (Binder GmbH, Tuttlingen, Germany) climate box was used for the long-term and accelerated stability test. The samples were stored in sealed containers, and their protection from light was ensured by covering the containers with a layer of foil. Evaluation of the chemical composition, physical stability and sensory properties, including color, odor and homogeneity were further investigated.

### 2.5. Evaluation of Physical Properties and Chemical Composition

The viscosity of the newly formulated teat dip solution was determined using the standard test method (STM), which was prepared according to the European Pharmacopoeia (Ph.Eur.) [15]. Viscosity measurements were performed at 20 °C utilizing a rotary NDJ-1 viscometer manufactured by COMECTA S.A. (Italy).

Three measurements were taken for each sample, and the average was calculated. Absolute viscosity was calculated using the following formula: η = k × α, where η—absolute viscosity; k—coefficient (selected according to the table in the viscosimeter instructions); and α—a number showing the value of the indicator.

The pH measurement was conducted with a laboratory InoLab pH 7310 pH-meter (Xylem Analytics Germany GmbH, Oberbayern, Germany) using a standard potentiometric test method, which was prepared according Ph.Eur. [15]. The centrifugation test of the final product was performed after a heating/cooling cycle and after 2 months keeping the dip at +45 °C. Briefly, 8 mL of experimental, newly formulated teat dip solution was added to the CLEARLine^®^ centrifuge tubes and centrifuged at a speed of 4500 rpm for 8 min. In order to evaluate the stability of the experimental new teat dip solution formula, separation of the aqueous phase and turbidity were observed. The centrifugation test was performed with an EBA 20 laboratory centrifuge (Andreas Hettich GmbH, Tuttlingen, Germany).

Odor, color and homogeneity were assessed using the Light Cabinet (Byko—Spectra Basic). The visual method was used for the evaluation of the homogeneity. Briefly, four samples of 0.02 g of the test product were prepared using two glass slides. The dip solution was placed on a single slide and pressed by the other slide firmly. Spots with a diameter of 2 cm were formed. When viewed through the light at a distance of 30 cm, at least three of the four samples are free of individual particles. If particles were observed in a larger number of samples, the product was considered to be non-homogeneous. 

For the odor testing, a small (0.5–2.0 g) sample was added to a chemical beaker, and after 15 min, an odor was detected organoleptically [15]. The evaluation was given by referring to the characteristic odor of the product.

Color determination was performed by a visual method [15]. The test procedure involved examining a thin layer of the product against a white background, and the rating was given by referring to the color characteristic for the product.

### 2.6. Antimicrobial Testing of the Prepared Teat Dip

The antimicrobial activity of the teat dip solution containing chlorhexidine was assessed using the dilution–neutralization method under experimental conditions according to the BS EN 1656:2019 and BS EN 1657:2016 standards [16]. For antimicrobial activity testing, the strains used were *Staphylococcus aureus* ATCC 6538, *Escherichia coli* ATCC 10536, *Streptococcus uberis* ATCC 19436, *Candida albicans* ATCC 10231 and *Aspergillus niger* ATCC 16404. The logarithm was used to express the indicator of product activity against the microorganisms used in the study. If the logarithm value obtained during the test exceeded 5 (log R > 5), the teat dip solution was considered as acting as a bactericidal substance. Antifungal (against the yeasts and fungi) activity was considered if the log exceeded 4, i.e., log R > 4. The logs were counted according to the formula: Log R = Log N_o_ − Log N_a_.

### 2.7. Statistical Analysis

The results were analyzed using statistical data analysis packages: SPSS 17.0 and Microsoft Office Excel 2010. Applying the paired-samples *t*-test, the significance of the differences in the results was determined, which is recorded when *p* < 0.05.

## 3. Results

During the study, the simulation of the teat dip with chlorhexidine was carried out according to the instructions in the methodology, and the selected ingredients were used (Table 2).

While optimizing and selecting the composition for the experimental solution, an in vivo study was performed. The formation of a simulated teat dip solution drop on the end of the teat and its retention time were evaluated, i.e., whether the teat dip solution properly covered (“closed”) the teat channel and its integrity, the formation of a protective film, visibility (the color was evaluated) and economy (the dripping after dipping was evaluated). The data are presented in Table 4.

During the study, it was found (Table 4) that the first experimental composition with a concentration of 0.5% thickener was insufficient, because immediately after dipping, the experimental product dripped intensively, covered the skin of the teat with a thin film, which was unstable due to heavy dripping, and did not form a drop at the end of the teat.

The third experimental composition of the new model of teat dip solution with a 1.2% amount of hydroxypropyl guar gum was not satisfactory (Table 4), because immediately after dipping, the experimental product dripped intensively and abundantly (4–6 drops per minute), the teat was covered with a too thick film, which resulted in a high yield of the product. The second experimental composition, when the hydroxypropyl guar gum concentration of 0.89% was used, which covered the teat with an even film, after a few minutes, a stable hanging drop was formed at the end of the teat, which did not fall off, and it lasted for about 40 min. The viscosity of the second EC was 1450–1500 mPa.s. All three experimental formulations used different amounts of dye, selected respectively, at 0.02–0.05–0.12%. After dipping, the color was clearly visible (red), and the color remained visible even after 50–60 min, but later, the intensity of the color remained practically unchanged in the second and third experimental compositions. So, depending on the dye concentration after 12 h, the color residues were visible only after using 0.05–0.12% of dye. After evaluating the data presented in Table 4, the optimal second experimental composition with a 0.89% amount of hydroxypropyl guar gum and 0.05% of dye was selected for further study.

In the second stage, stability studies of the optimal teat dip solution composition were carried out, i.e., the experiments were carried out in order to evaluate the possible changes in the composition’s visual/sensual properties (color, odor and homogeneity), pH, and viscosity parameters. Three batches of the experimental composition of the modeled optimal composition were produced for the stability study.

The cosistency of the modeled composition was homogene-ous, the color remained bright red as its odor characteristic being retained, and all these properties did not changed on the 6th and 12th months after production date (Table 5). During the follow-up period, after 24 months, this product changed visually, while delamination and turbidity of the modeled compositions were found during preliminary tests. 

The results of the stability study showed that the pH value of the modeled teat dip solution remained stable in the long-term program, unchanged over 12 months (Table 5), while the pH of EC in the optimal composition was within 5.8–6.1 (*p* > 0.05). The pH was close between different batches and remained stable in the modeled product for 12 months (*p* = 0.83). The statistical data analysis showed that there were differences after 24 months, where the pH value was 5.2–5.3, but these were not significantly different (*p* = 0.25).

The viscosity readings of the modeled teat dip solution 24 h after production in different series of samples were, on average, within the limits of 1478 ± 18 mPa.s (Table 5), and no differences were found between the teat dip solutions series (*p* > 0.05). Viscosity readings during storage varied evenly, although they were not significantly different (*p* = 0.72), and the Pearson correlation coefficient was r = 0.96–0.99. The long-term stability program determined the viscosity changes after 12 months (2.6% decrease) and 24 months (3.9%), but these were not significantly different (*p* = 0.48).

The color, homogeneity and odor remained unchanged during the accelerated stability program. After the centrifugation tests, the investigated products did not change, they remained stable during the entire test, and the product phases did not separate (Table 6).

During storage of the samples for the new modeled teat dip solution at a +45 °C temperature for 2 months, no significant (*p* = 0.75) deviation in the pH value from the initial value was found (Table 6). Furthermore, it can be affirmed that the pH value of the samples remained constant during the heating/cooling cycles, and the readings exhibited stability (*p* = 0.25). The stressful experiment conditions did not cause a change between the batches, with the pH value remaining strongly correlated between the batches (r = 0.96–0.98).

The third task in evaluating the new modeled teat dip solution was to check its bactericidal, anti-yeast and antifungal activities. The test method described in the standard evaluated the effectiveness of the teat dip solution in order to reduce the number of viable microorganisms used in the test. The study showed that the 50% and 80% teat dip solution concentrations were bactericidally active against all the bacteria tested strains (log R > 5, Table 7). After diluting the teat dip solution to 10%, it lost effectiveness. However, a 10% teat dip solution was effective against Gram-negative bacteria (*E. coli*). The 80% and 50% teat dip solution concentrations inhibited the growth of reference *Candida albicans* yeast strains, whereas the 10% teat dip solution had no anti-yeast effect. The results are presented in Table 7.

## 4. Discussion

During the development of the teat dip solution, the first task was to model a product that would properly cover the skin of the teat, would last for the required time and would properly color the skin of the teat after dipping.

Following the identification of the suitable hydroxypropyl guar thickener (guar gum), CI 16255 dye and chlorhexidine digluconate concentrations, a new teat dip solution was created that also contained glycerol, polysorbate 80, isopropyl alcohol, *Mentha arvensis* herb oil and *Aesculus hippocastanum* dry extract. The teat dip solution was modulated for post-milk udder hygiene with a viscosity of 1478 ± 18 mPa.s. When such a viscosity is introduced for in vivo investigation, the product spread quickly on the skin and achieved the desired consistency, forming the drop at the end of the teat, dying the skin of the teat and holding the consistency during the storage period. The 5.9 ± 0.18 pH of the solution was very close to natural pH for the teat skin. While modeling the product, the main ingredients provided viscosity, color and the antimicrobial effect, while the other ingredients performed technological and auxiliary roles.

In biocidal teat dip solutions, the most important parameters are the disinfecting effect and sufficient viscosity. Product viscosity is an important physical parameter that reflects product quality [17,18]. Therefore, we paid great attention to obtaining the suitable viscosity in our study. As other researchers indicate, guar gum performs the role not only of a thickener but also forms a protective film, performs an emulsifying effect and stabilizes the product during development [17]. While evaluating the literature data, it was found that the amount of guar gum used in the product formulations varies from 0.1 to 0.3% [19]. We have used the hydroxypropyl guar polymer, C3H8O2 x (isomer), which is a purified non-ionic derivative of guar gum. It works as an excellent thickener in cosmetic formulations containing a high polar solvent content. It also works as an O/W emulsion stabilizer and is particularly useful in stabilizing emulsions containing alcohol [20]. Thanks to its polymeric structure, this guar derivative is compatible with the skin, leaving a soft feeling after application [19]. During the research, we chose hydroxypropyl guar gum, which forms the viscosity of the new product. Hydroxypropyl guar gum is sufficient in the product at 0.89%. This thickener concentration kept the formulated product stable statistically reliably throughout the study period, i.e., monitoring for 1 year after manufacture. If it is stored for a longer period (after 24 months), the viscosity reduces, the product changes its visual appearance and the pH reduces to 5.2 ± 0.05.

Developing new products, it is very important that products are not hazardous to animals, human health or the environment. Therefore, attention was drawn to the classification of thickener and antimicrobial substance according to EU legislation [21]. The classification of the chemical mixture is a particularly important factor in teat dip solutions. According to EU legislation, the hydroxypropyl guar gum polymer in our product does not pose a risk to animal health or the environment. In addition, the chlorhexidine withdrawal period is not applied for the diary animals.

While modeling the product, a red dye was selected, the concentration of which in the product met the raised requirements; the color remained bright and stable until the next milking, and its optimal concentration in the modeled product was 0.05%.

As Kumar et al. (2014) [22] indicate, the selection of auxiliary materials is important for the production of local effect products. Each ingredient has its function in the product. The proper selection of the excipients can effectively form stable systems. Auxiliary substances used as supplements in the product also protected and preserved the skin of the teat. It is indicated in the literature that glycerol moisturizes the stratum corneum of the skin, increases elasticity, protects against external irritants and improves wound healing. There is indicative data that glycerol can also have an antimicrobial effect [23,24]. Plant extracts are widely used in the world and are of great importance not only for human health but also animal health [25]. In dairy farming, it is important that phytotherapeutic preparations have no side effects, no shelf life and do not cause bacterial resistance. The mint essential oil was chosen for our products, as it has a specific deodorizing odor and ensures a cooling, stimulating, warming and refreshing effect [26]. It interacts with cold-sensitive TRPM8 receptors in the skin, which are responsible for sensing the cooling effect [27]. Menthol has antiseptic, antiviral features that stop the growth of microorganisms and fungi [28]. Another herbal preparation that we have used is chestnut extract. According to the researchers, horse chestnut extract improves skin elasticity, helps with skin diseases and, in addition, β-escin determines the anti-edematous and vasoprotective effect of horse chestnut fruits, which inhibits inflammation, reduces swelling and relieves pain [29,30].

While modeling our product, isopropanol and polysorbate 20 were chosen as technological materials which mix well with other ingredients. This feature of isopropyl alcohol allows it to be used for dissolving many organic compounds, and it is frequently used as a solvent in pharmaceuticals and cosmetics. This alcohol is also used as a disinfectant because it prevents the proliferation and growth of microorganisms [31]. Technological substances, such as citric acid and sodium hydroxide, allowed us to model the required pH of the product, which is close to the natural pH of cattle skin. It is important to ensure the correct pH so that the product would not irritate the skin. Therefore, this optimally formulated moisturizer has the right pH to moisturize the skin of the teat. 

When evaluating the quality of biocidal products, the stability studies of physical and chemical parameters are important. The testing of stability indicates the quality sufficiency of manufactured products throughout the shelf life and the product’s retaining of its original chemical and physical properties. If the product is unstable, its active substances can change, and the product itself can lose its disinfecting characteristics. When the viscosity in the teat dipping solution is reduced, no film is formed on the teat, which would ensure the disinfectant properties while retaining the active substances at the destination area.

Chang R.K. and colleagues, in an article dating back to 2013 [32], indicated the recommended tests for products which are intended to be used on the skin and which can be useful in assessing the quality of products (testing of organoleptic features, assessing visual homogeneity, determining pH, assessing product consistency etc.). Most of these tests are not regulated as mandatory but are recommended to verify and confirm the quality of the modeled product. To register a biocidal product in the European Union, it is it is imperative to provide the findings of stability studies, which determine the shelf life of the product [18]. We evaluated the parameters defining the stability of the developed products in long-term and accelerated stability programs. The product’s physical integrity, stability and chemical stability were tested. After evaluating the test results, it was found that the teat dip was stable at low, high and room temperatures. The viscosity and pH results of the samples were also within the initial limits (*p* > 0.05), i.e., stable. The results of the study show that the change in storage conditions does not have a significant impact on the physical and chemical properties of the newly developed teat dipping solution as well as the product’s appearance and organoleptic properties.

A disinfectant is a necessary component in udder hygiene, and we have chosen chlorhexidine. This substance is one of the most widely used biocides in veterinary and animal husbandry. Chlorhexidine is a clinically important antiseptic, disinfectant and preservative. It is a potent membrane-active agent against bacteria, and it inhibits outgrowth but not the germination of bacterial spores, although it is not sporicidal. It shows a high activity against wild-type and outer membrane mutants of *E. coli* [33]. Chlorhexidine digluconate (CHX) has been known as an antiseptic since the early 1950s for clinical use. It is widely used in medicine and veterinary as a broad-spectrum antimicrobial substance that destroys cell membranes. Vianna et al. [34] demonstrated the antimicrobial efficacy of CHD in a study. Analyzing in vitro the antimicrobial activity of different CHD concentrations (0.2%, 1% and 2%) against pathogens, it was found that 2% CHD (in gel or solution form) was bactericidal active against *S. aureus* and *C. albicans* in 15 s of exposure; meanwhile, the bactericidal effect on the *E. faecalis* culture was fixed within 1 min. Also, Gilbert, Moore (2005) [35] and Brookes et al. (2020) [36] additionally extended the analysis of the antimicrobial activity of CHD and found out that the antifungal effect of CHX is also related to the prevention of biofilm formation: it is not only destined to damage the microbial structure or the cell membrane. This has been proven in studies with *C. albicans* cultures. Evans et al. (2009) [37] and Babickaite et al. (2016) [38] were investigating the CHX soak and gel in different concentrations. They pointed out that for the expected result, the appropriate chlorhexidine concentration and contact time are the most important factors.

In the research carried out by Riffon et al. (2001) [39] and Fitzpatrik et al. (2021) [40], it was indicated that within the European Union member states, there must be a common standard to evaluate teat disinfectant products. This European standard (EN), which is known as BS EN 1656, can be used to compare a range of disinfectants. We conducted a study under these standards with a modeled product as well. Our newly developed teat dip solution demonstrated a bactericidal effect. It was confirmed by an in vitro study against reference microbial isolates. Our research, according to the BS EN 165601:2019 or BS EN 1657:2016 standards, showed a bactericidal activity at 80% and 50% concentration to Gram-positive, Gram-negative bacteria and *C. albicans*.

Although the antifungal effect was not detected in the developed teat dip solution, it could be treated as a promising tool for the prevention of cow udder infections, which is known to be caused by multiple species of bacteria and only in some cases by fungi.

## 5. Conclusions

The modeled product has a homogeneous appearance, a red color and a specific odor of mint essential oil and lasted for the required time; after dipping, it properly covered the skin of the teats, had the desired consistency, did not drip and formed a drop at the end of the teat. 

The desired features of the product were modeled using a chlorhexidine digluconate 20% solution of 2.5 g, the viscosity-forming polymer hydroxypropyl guar gum (0.89%) and the dye (CI 162551) (0.05 g). Product viscosity was 1450–1500 mPa.s, and the pH value was 5.8–6.1.

The quality of the modeled product did not change for 12 months in the long-term stability program: the consistency was homogeneous, the color remained bright, the odor was characteristic of the product and the pH value was stable. Difficult experimental conditions did not have a statistically significant effect on the pH value: the pH value of the modeled product fluctuated within stable product-specific limits.

Both 80% and 50% concentrations of the simulated teat dip solution inhibited the growth of *S. aureus*, *S. uberis*, *E. coli* and *C. albicans* strains. *Aspergillus niger* was not susceptible to the tested product; therefore, improved formulas for the teat protection from the pathogens should be explored further.

## Figures and Tables

**Table 1 vetsci-10-00510-t001:** Short study design for development and testing of the teat dip solution.

**1.** Development of experimental compositions of teat dip solution			
**2.** Evaluation and selection of experimental formulations (in vivo)			
**3.** Evaluation of the stability of the physico-chemical parameters of the selected composition			
Viscosity	pH	Appearance of teat dip solution	Stability
**4.** Study of the antibacterial and antifungal effects of the selected composition			

**Table 2 vetsci-10-00510-t002:** Composition of the developed new model teat dip solution (100 g product).

Product Composition	CAS No.	Molecular Mass	Manufacturer	Quantity, g		Function
Chlorhexidine digluconate (D-gluconic acid, compound with N,N″-bis(4-chlorophenyl)-3,12-diimino-2,4,11,13-tetraazatetradecanediamidine (2:1), C22H30Cl2N10 × 2C6H12O7	18472-51-0	M = 897.77 g/mol	Medichem, S.A., 08970 Sant Joan Despi, Spain	0.50		Active substance (antimicrobial properties)
Hydroxypropyl guar, C3H8O2 × (isomer)	39421-75-5	M = 536.44 g/mol	Lamberti SpA, 21013 Gallarate, Italy	No.1	0.50	Stabilizer, thickener, viscosity modifier
				No.2	0.89	
				No.3	1.20	
Dye CI 16255 (Trisodium (8Z)-7-oxo-8-[(4-sulfonatonaphthalen-1-yl)hydrazinylidene]naphthalene-1,3-disulfonate), C20H11N2Na3O10S3	2611-82-7	M = 604.46 g/mol	Neelikon Food Dyes & Chemicals Ltd., 402116 Dhatav, Maharashtra, India	No.1	0.02	Colorant
				No.2	0.05	
				No.3	0.12	
Glycerol (1,2,3-propantriol), C3H5(OH)3	56-81-5	M = 92.09 g/mol	Aarhus Karshams Sweden AB, 37431 Karlshamn, Sweden	5.10		Skin emollient, moisturizer
*Mentha arvensis* herb oil (L-Menthol ≥ 35.0%, Menthone (17.0–25.0%, Cineole ≤ 1.0%, Isomenthone ≥ 13.0%, Menthylacetate 2.0–6.0%)	90063-97-1	The oil is blend of various substances *	Düllberg Konzentra GmbH & Co. KG, 22335 Hamburg, Germany	0.10		Cooling, antiseptic, perfuming function
*Aesculus hippocastanum* dry extract from seeds (Aescin 18.0–22.0%) (Extract Ratio 7:1) (Extraction solvent: Ethanol/Water (40/60 *v*/*v*)	8053-39-2	The extract is blend of various substances *	Dorana Naturae, 81108 Bratislava, Slovakia	0.01		Antioxidant, skin conditioning and protecting effect
Polysorbate 80, C64H124O26	9005-65-6	M = 1310.00 g/mol	Oleon NV, 9940 Evergem, Belgium	3.00		Emulsifier
Isopropyl alcohol, C3H7OH	67-63-0	M = 60.10 g/mol	Rebain|International NL, 3059 LM Rotterdam, The Netherlands	5.00		Solvent
Sodium hydroxide, NaOH	1310-72-2	M = 92.09 g/mol	Stanchem Sp. z o.o., 21025 Niemce, Poland	q.s.		pH adjustment
Citric acid monohydrate, C6H8O7 × H2O	5949-29-1	M = 210.14 g/mol	Reachem s.r.o, 83103 Bratislava, Slovakia	q.s.		pH adjustment
Purified water, H2O	7732-18-5	M = 18.01 g/mol		Up to 100 g		Solvent

P.S.—* no molecular mass.

**Table 3 vetsci-10-00510-t003:** Evaluation criteria of experimental teat dip solutions in vivo.

Parameters	Evaluation Criteria
Color	
1 point	After dipping, the color is hard to see. After 60 min, no color is visible.
2 points	After dipping, the color is bright and lasts well for 10 min. After 60 min, the color is hard to see.
3 points	After dipping, the color is bright and lasts well for 10 min. After 60 min, the color did not change. After 12 h, the remains of the color are not visible.
4 points	After dipping, the color is bright and lasts well for 10 min. After 60 min, the color did not change. After 12 h, the color residues are visible only on part of the teats.
5 points	After dipping, the color is bright and lasts well for 10 min. After 60 min, the color did not change. After 12 h the remains of the color are visible.
Dripping immediately after dipping	
1 point	After dipping, the dripping is intense (more than 6 drops in the first minute).
2 points	After dipping, the dripping is intense (4–6 drops in the first minute).
3 points	After dipping, the dripping is moderate (2–4 drops in the first minute).
4 points	After dipping, the dripping is slow (no more than 2 drops in the first minute).
5 points	No dripping after dipping (no more than one drop in the first minute).
Formation of a drop on the teat end	
1 point	The drop does not form; dripping is too intense.
2 points	After 5 minutes, an elongated drop is formed that does not fall off. After 30 min, the hanging drop is no longer visible.
3 points	After 5 minutes, an elongated drop is formed that does not fall off. After 60 min, the hanging drop is no longer visible.
4 points	After a few (2–3) minutes, a stable hanging drop is formed, which lasts for about 40 min. After 60 min, the hanging drop is no longer visible.
5 points	After a few (2–3) minutes, a stable hanging drop is formed. After 60 min, the hanging drop is visible.
Teat covered with a film	
1 point	Covers with a very thick film, high product yield.
2 points	Covers with a thick film, high product yield.
3 points	Covers with a thin film, which becomes even thinner and unstable due to heavy dripping.
4 points	Covers the skin of the teat with a sufficiently even film, in a sufficiently even layer.
5 points	Covers the skin of the teat with an even film, in an even layer.
Evenness of teat coverage	
1 point	Covers unevenly.
2 points	Covers satisfactory evenly.
3 points	Covers averagely evenly.
4 points	Covers evenly enough.
5 points	Covers evenly.
General evaluation of experimental formulations of the teat dip solution	
	Total points

**Table 4 vetsci-10-00510-t004:** Evaluation study of different experimental teat dip solutions in vivo.

Parameters	Experimental Formulas and the Results of the Evaluation of the Tested Parameters		
	Composition 1 (Hydroxypropyl Guar Gum 0.5%, Dye 0.02%)	Composition 2 (Hydroxypropyl Guar Gum 0.89%, Dye 0.05%)	Composition 3 (Hydroxypropyl Guar Gum 1.2%, Dye 0.12%)
Color			
Evaluation: (not visible—1, clearly visible—5)	3 points After dipping, the color is bright and lasts well for 10 min. After 60 min, the color did not change. After 12 h, the color disappeared.	5 points After dipping, the color is bright and lasts well for 10 min. After 60 min, the color did not change. After 12 h, the remains of the color are visible.	5 points After dipping, the color is bright and lasts well for 10 min. After 60 min, the color did not change. After 12 h, the remains of the color are visible.
Dripping immediately after dipping			
Evaluation: (intense dripping—1, no dripping—5)	1 point After dipping, the dripping is intense (about two minutes).	3 points After dipping, the dripping is moderate (3–4 drops in the first minute).	2 points After dipping, the dripping is intense (4–6 drops in the first minute).
Formation of a drop on the teat end			
Evaluation: (not properly formed—1, properly formed—5)	1 point The drop does not form; dripping is too intense.	4 points After a few minutes, a stable hanging drop is formed, which lasts for about 40 min. After 60 min, the hanging drop is no longer visible.	3 points After five minutes, an elongated drop is formed that does not fall off. After 60 min, the hanging drop is no longer visible.
Teat covered with a film			
Evaluation: (covered with a thick film—1, covered with a thin film—5)	3 points Covers with a thin film, which becomes even thinner and unstable due to heavy dripping.	5 points Covers the skin of the teat with an even film in an even layer.	2 points Covers with a thick film, high product yield.
Evenness of teat coverage			
Evaluation: (covers unevenly—1, covers evenly—5)	4 points Covers evenly enough	4 points Covers evenly enough	4 points Covers evenly enough
General evaluation of experimental formulations of the teat dip solution			
	12 points	21 points	16 points

**Table 5 vetsci-10-00510-t005:** Long-term stability test results.

Evaluation Period	Homogeneity	Color	Odor	pH	Viscosity, mPa.s
After 24 h	Homogeneous	Red	Characteristic mint essential oil	5.9 ± 0.18	1478 ± 18
After 6 months	Homogeneous	Red	Characteristic mint essential oil	5.9 ± 0.18	1460 ± 14
After 12 months	Homogeneous	Red	Characteristic mint essential oil	5.9 ± 0.18	1435 ± 15
After 24 months	Not homogeneous, dark precipitate observed	Red	Characteristic mint essential oil	5.2 ± 0.05	1420 ± 10

**Table 6 vetsci-10-00510-t006:** Results of the accelerated stability tests.

Evaluation Period	Homogeneity	Color	Odor	pH	Centrifugation, Speed 4500 rpm, Time 8 min
After manufacturing	Homogenic	Red	Characteristic mint essential oil	5.9 ± 0.2	Not layered, stable
Heating/cooling cycles	Homogenic	Red	Characteristic mint essential oil	5.9 ± 0.1	Not layered, stable
After 2 months	Homogenic	Red	Characteristic mint essential oil	5.9 ± 0.2	Not layered, stable

**Table 7 vetsci-10-00510-t007:** Teat dip solutions containing chlorhexidine (80%, 50%, 10%) antimicrobial activity (log R) according to the UNI EN 1656 and UNI EN 1657 standards.

Test Microorganisms	Test Suspension	Results		
		80%	50%	10%
*S. aureus* ATCC 6538	10^−6^: >330–>330 * 10^−7^: 39–42 * N: 4.05 × 10^8^ N_o_: 4.05 × 10^7^ log N_o_: 7.61	Active log R > 5.46 V_c_: <14–<14 N_a_ < 140 log N_a_ < 2.15	Active log R > 5.46 V_c_: <14–<14 N_a_ < 140 log N_a_ < 2.15	Not active log R = 4.35 V_c_: 189–171 N_a_ = 1800 log N_a_ = 3.26
*S. uberis* ATCC 19436	10^−6^: >330–>330 * 10^−7^: 38–48 * N: 4.30 × 10^8^ N_o_: 4.30 × 10^7^ log N_o_: 7.63	Active log R > 5.48 V_c_: <14–<14 N_a_ < 140 log N_a_ < 2.15	Active log R > 5.48 V_c_: <14–<14 N_a_ < 140 log N_a_ < 2.15	Not active log R < 4.11 V_c_: >330–>330 N_a_ > 3300 log N_a_ > 3.52
*E. coli* ATCC 10536	10^−6^: >330–>330 * 10^−7^: 42–46 * N: 4.40 × 10^8^ N_o_: 4.40 × 10^7^ log N_o_: 7.67	Active log R > 5.49 V_c_: <14–<14 N_a_ <140 log N_a_ < 2.15	Active log R > 5.49 V_c_: <14–<14 N_a_ < 140 log N_a_ < 2.15	Active log R > 5.44 V_c_: 18–<14 N_a_ < 160 log N_a_ < 2.2
*C. albicans* ATCC 10231	10^−5^: >330–>330 * 10^−6^: 38–44 * N: 4.10 × 10^7^ N_o_: 4.10 × 10^6^ log N_o_: 6.61	Active log R > 4.46 V_c_: <14–<14 N_a_ < 140 log N_a_ < 2.15	Active log R > 4.46 V_c_: <14–<14 N_a_ < 140 log N_a_ < 2.15	Not active log R < 3.09 V_c_: >330–>330 N_a_ > 3300 log N_a_ > 3.52
*Aspergillus niger* ATCC 16404	10^−5^: >165–>165 * 10^−6^: 28–32 * N: 3.00 × 10^7^ N_o_: 3.00 × 10^8^ log N_o_: 6.48	Not active log R < 3.26 V_c_: >165–>165 N_a_ > 1650 log N_a_ > 3.22	Not active log R < 3.26 V_c_: >165–>165 N_a_ > 1650 log N_a_ > 3.22	Not active log R < 3.26 V_c_: >165–>165 N_a_ > 1650 log N_a_ > 3.22

P.S.—N_o_ = number of CFU/mL in the test mixture; R = reduction in viability; N_a_ = number of CFU/mL of the test mixture; V_c_ = viable count; N = number of CFU/mL of the test suspension; * number of colonies (CFU) in test suspension dilutions.

## Data Availability

The datasets generated and/or analyzed during the current study are available from the corresponding author upon reasonable request.

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
