# Peer review of "A Study on the Stability and Antimicrobial Efficacy of a Newly Modeled Teat Dip Solution Containing Chlorhexidine"

_vetsci, 2023, doi:10.3390/vetsci10080510_

Round 1

Reviewer 1 Report

The study present the differnt stages from conception to test a new post-milking teat dip solution.

There are already a lot of those products but it would be interesting from a local or regional, or industrial point of view.

I have to say that I stopped there my lecture. You can find thereafter the first corrections proposed.

-Avoid plagiarism  (some sentences are exactly the same that references, l39-40, 41-42, 53-56), and references to "the researchers working" (line 44).

-Take only references useful for the reader to understand the thematic and its positioning within the scientific context (reference 5).

-Carefully reread your text to avoir repetition (sentence lines 76-77)

- Don't forget the objectives and rapid description of the study at the end of the introduction part.

Material and Methods

-Present here the characteristics/advantages of the teat dip that you want to : Ideal viscosity (in front of retention and cleanliness), color (characteristics in front of the usage (compatible with oral consumption in human and calves for example)

-Present completely the products (Phytoproducts are not presented with their extraction characteristics and what part of the plant extract)

- Propose a figure with the different steps of your procedures of elaboration, with the different timing for tests of stability: (Line 155 : 2 months at 45°C is not described ). Also, if stability test with sealed containers is important, we don't have any idea of the evolution of the product at air. A teat dip stay open sometimes very long time in small scale herds!

-It seem better to evaluate the colour with wavelentgth

- Don't forget in vivo description of the study!

Results:

Sentence L192-194 = discussion or introduction

The authors have omitted any description of the in vivo methods employed in the material and methods section. However, it is worth noting that they have conducted tests on teat skin, at least , using different formulations (lines 205-208). Addressing this gap would be a significant improvement before submitting their study for review.

Table 3: there is nowhere description of your in vivo testing!!!

Author Response

Dear Reviewer, 

We are very thankful for your time and efforts to read our manuscript and to provide suggestions and for possibility to improve it.

The authors have improved the methods, results and conclusion parts based on your comments.

All response to your comments is in the added file (Please see the attachment "Response to the Reviewer No.1"), all changes are visible in added corrected manuscript with "track changes".

The authors also note that changes were made in manuscript according suggestions of other reviewers as well as according Editor suggestion to reduce the repetition rate and improve of English language.

On behalf of authors,

Modestas Kybartas

Lithuanian University of Health Sciences

Reviewer 2 Report

Please find my comments in the attached file

There is ample room for enhancement throughout the manuscript. Several grammatical errors and sentence structures require attention and improvement. 

Author Response

Dear Reviewer,

We are very thankful for your time and efforts to read our manuscript and to provide suggestions for possibility to improve it.

The authors have improved the methods, results and conclusion parts based on your comments and added relevant references.

 All response to your comments is in the added file (Please see the attachment “Response to the Reviewer No.2”), all changes are visible in added corrected manuscript with "track changes".

The authors also note that changes were made in manuscript according to the suggestions of the other reviewers and also according to the Editor remarks with the aim to to reduce the repetition rate and to improve English.

On behalf of authors

Modestas Kybartas

Lithuanian University of Health Sciences

Round 2

Reviewer 1 Report

Some sentences stay unclear;

Author Response

We are very thankful for your time and efforts to read our manuscript again and to provide suggestions and for possibility to improve it.

The authors have improved the manuscript based on your comments.

All response to your comments is in the added file (Response to the Reviewer No.1 after minor revision), all changes after second revision are marked in yellow color in added in the corrected manuscript with "track changes".

The authors also note that changes were made to improve English.

On behalf of authors,

Modestas Kybartas

Lithuanian University of Health Sciences

Reviewer 2 Report

I am happy for the manuscript to be accepted after the revision.

Author Response

We are very thankful for your time and efforts. We are happy that you approved all changes we have made according your previous comments.

On behalf of authors,
Modestas Kybartas
Lithuanian University of Health Sciences